# The Silent Threat: Antimicrobial-Resistant Pathogens in Food-Producing Animals and Their Impact on Public Health

**DOI:** 10.3390/microorganisms11092127

**Published:** 2023-08-22

**Authors:** Ayidh M. Almansour, Meshari A. Alhadlaq, Khaloud O. Alzahrani, Lenah E. Mukhtar, Abdulmohsen L. Alharbi, Sulaiman M. Alajel

**Affiliations:** 1Molecular Biology Division, Reference Laboratory for Microbiology, Executive Department of Reference Laboratories, Research and Laboratories Sector, Saudi Food and Drug Authority (SFDA), Riyadh 11671, Saudi Arabia; mahadlaq@sfda.gov.sa (M.A.A.); kozahrani@sfda.gov.sa (K.O.A.); alharbi@sfda.gov.sa (A.L.A.); 2Antimicrobial Resistance Division, Reference Laboratory for Microbiology, Executive Department of Reference Laboratories, Research and Laboratories Sector, Saudi Food and Drug Authority (SFDA), Riyadh 11671, Saudi Arabia; lemukhtar@sfda.gov.sa; 3Reference Laboratory for Microbiology, Executive Department of Reference Laboratories, Research and Laboratories Sector, Saudi Food and Drug Authority (SFDA), Riyadh 11671, Saudi Arabia; smotibi@sfda.gov.sa

**Keywords:** antibiotics, livestock, One Health approach, multidrug resistance, foodborne, pathogens, healthcare costs

## Abstract

The emergence of antimicrobial resistance (AMR) is a global health problem without geographic boundaries. This increases the risk of complications and, thus, makes it harder to treat infections, which can result in higher healthcare costs and a greater number of deaths. Antimicrobials are often used to treat infections from pathogens in food-producing animals, making them a potential source of AMR. Overuse and misuse of these drugs in animal agriculture can lead to the development of AMR bacteria, which can then be transmitted to humans through contaminated food or direct contact. It is therefore essential to take multifaceted, comprehensive, and integrated measures, following the One Health approach. To address this issue, many countries have implemented regulations to limit antimicrobial use. To our knowledge, there are previous studies based on AMR in food-producing animals; however, this paper adds novelty related to the AMR pathogens in livestock, as we include the recent publications of this field worldwide. In this work, we aim to describe the most critical and high-risk AMR pathogens among food-producing animals, as a worldwide health problem. We also focus on the dissemination of AMR genes in livestock, as well as its consequences in animals and humans, and future strategies to tackle this threat.

## 1. Introduction

Livestock such as poultry, cows, and cattle are widely produced around the world, primarily for food consumption and financial gain [1]. Livestock plays a significant role in our society, both in terms of its importance for food production and its economic value.

In spite of this, there are also challenges associated with livestock farming, particularly regarding antimicrobial resistance (AMR) [2,3]. One significant contributor to the development and spread of this worldwide problem is the misuse of antibiotics in livestock farming [4,5]. Antibiotics are often used in livestock production for promoting growth and preventing or treating infections. In developing countries, this practice involves administering low doses of antibiotics to healthy animals over prolonged periods, promoting selective pressure on bacteria and, consequently, the survival and growth of resistant strains. The constant exposure to antibiotics leads to the emergence of resistant strains.

AMR occurs when bacteria become less susceptible (i.e., resistant) and develop resistance to the drugs used to treat them (antibiotics). This resistance makes infections caused by these microorganisms more difficult to treat and can lead to increased morbidity, mortality, and healthcare costs. AMR is influenced by a range of factors that affect human, animal, and environmental health. These factors include clinical, biological, social, political, economic, and environmental issues that can accelerate the emergence and spread of AMR [6]. One critical aspect is the failure to adhere to the withdrawal period after administering antibiotics. This AMR phenomenon increases the likelihood of the spread of diseases, treatment failure, severe illness, and in some cases, fatalities [7,8]. Particularly, the fast adaptation and dissemination of resistant microbes affect the treatment of common infections, such as urinary tract infections (UTIs), as well as more severe and potentially fatal illnesses, including tuberculosis, bacteremia, and pneumonia [9,10]. 

Several microorganisms have been identified frequently in different environments (healthcare settings, agriculture, or community settings). *Staphylococcus* species, *Salmonella* species, *Enterococcus* species, *Campylobacter species*, and *Enterobacteriaceae* [11,12] are some of them, which will be explored in detail in the next section of this review article. Particularly, food-producing animals have been identified as an important reservoir for the transmission of antibiotic-resistant *Enterobacteriaceae*, and their potential impact on human health has received attention among the global scientific community [13].

Along this line, AMR has become a major challenge for public health worldwide [13]. The continued use of antibiotics over a prolonged period has created a force that only allows certain bacteria to survive, leading to these bacteria becoming resistant to antibiotics. According to current estimates, suboptimal prescribing of antibiotics occurs in up to 50% of cases, resulting from a range of factors, such as incorrect dosing, inappropriate selection of antibiotics, inadequate treatment duration, or even erroneous prescriptions for non-bacterial conditions [10].

This public health problem can also have a significant economic impact, as well as a severe effect on national healthcare systems. It can reduce productivity due to prolonged hospital stays and the need for more expensive and intensive care treatments. Previous studies have been conducted on AMR in food-producing animals. This review paper aims to provide an updated and comprehensive overview of the most critical AMR pathogens among food-producing animals and the widespread dissemination of AMR genes in livestock, as well as to discuss the potential consequences for both animal and human health and propose future strategies to combat this threat.

## 2. The Impact of Antibiotic Use in Animal Farming

To our knowledge, antibiotics are commonly administered in animal agriculture to enhance their health and productivity [14]. A recent study by Van et al. [15] highlighted that about 80% of livestock raised for food production receive treatments, including antibiotics, at certain stages. Merely 10% of antibiotics are utilized for treating infections, with the majority being employed for disease prevention and fostering growth [16]. Around half of the antibiotics utilized in the United States (US) are administered to animals in the agriculture field, mainly involving penicillin and tetracycline. In other words, a significant portion of antibiotics given to animals is not solely for therapeutic purposes, but is also used to enhance animal growth and prevent diseases [17]. 

AMR bacteria can then be transmitted to humans through the consumption of contaminated animal-derived food products. Moreover, this practice is also associated with the dissemination of plasmids and other mobile genetic elements, which can be transferred to humans, limiting the effectiveness of antibiotics [17]. 

A study conducted in seven European countries revealed a significant connection between the occurrence of AMR strains of *Escherichia coli* (*E. coli*) in livestock and the usage of particular categories of antibiotics [18]. According to other research performed by Lee et al. [19], the characteristics of *E. coli* found in animals were the same as those found in patients with UTIs. Moreover, Jans et al. [20] detected drug-resistant *Staphylococcus aureus* bacteria (frequently associated with humans and livestock) among milk products across western and eastern Africa. Between 2008 and 2011, in countries where cholera outbreaks were occurring, different authors reported that *Vibrio cholerae* developed resistance to co-trimoxazole, a common antibiotic used to treat cholera [21]. According to a study conducted across 18 African countries, the primary medication used to treat *Shigella* diarrhea is no longer effective in 78% of the 451 tested isolates [22]. Several antimicrobial agents, such as ampicillin, gentamicin, and erythromycin, used for livestock production are the same as those used in human medicine. These drugs are categorized as “critically important antimicrobials”.

In developed countries, the fight against AMR is still hindered by a lack of cohesive strategies, despite having strong laws and policies in place. Unfortunately, the use of antibiotics in animal rearing to meet this demand also contributes to an alarming increase in this global issue. 

Due to the combination of overuse of antibiotics and limited innovation in their development, the world is being brought back to a time when bacterial infections posed a significant threat. In addition, the process of discovering and developing new antibiotics is long and expensive, and many pharmaceutical companies have been reluctant to invest in this area. This fact has led to a significant decline in the number of new antibiotics being developed in recent years; therefore, it is important to raise awareness about the importance of antibiotics and the threat of multi-drug resistant (MDR) bacteria to encourage consumers, healthcare providers, and policymakers to prioritize the development of new antimicrobial agents [23]. 

## 3. Drivers of AMR in Food-Producing Animals

Some environmental factors also play a significant role in the development and spread of antibiotic-resistant bacteria (ARB) from food-producing animals to humans. One particular way is through the consumption of contaminated food products, such as meat, milk, and eggs [8,24,25]. Livestock feed, which is commonly used as fertilizer in agriculture, can contain large amounts of antibiotic-resistant bacteria and their genes [26]. Moreover, exposure to environments inhabited by ARB or direct contact with ARB by individuals working in agriculture is another possible mode of transmission [8]. 

Crowded and unsanitary living conditions in intensive animal farming operations provide a perfect breeding ground for bacteria to thrive, leading to an increased likelihood of bacterial infections; thus, the use of antibiotics in these conditions only further exacerbates the problem by promoting the growth of resistant bacteria. Dairy products made from livestock may contain antibiotics due to their use in manufacturing and preservation. Antibiotics can contaminate food products through air or water pollution during production, processing, and transportation. Moreover, animals indirectly transfer antibiotics to dairy products by consuming feed that contains antibiotics.

A recent systematic review published by Larsson and Flach [7] in *Nature* showed the restriction of the use of antibiotics in food-producing animals and its associations with AMR in human beings. In that review, a meta-analysis of 1422 studies revealed a distinct increase in ARGs among food-producing animals in countries with high levels of antibiotic use in food-producing animals compared to those with low use. The study also found that reducing antibiotic use in food-producing animals was associated with a reduction in AMR genes [7].

Overall, the drivers of AMR in food-producing animals are multifactorial; therefore, reducing the use of antibiotics in these animals, improving their living conditions, and improving the management of manure can all play a role in mitigating the spread of AMR in these animals.

## 4. Priority Pathogens in Food-Producing Animals

The gastrointestinal (GI) tract of humans and animals is inhabited predominately by Gram-negative and Gram-positive bacteria. *Enterobacteriaceae* is a family of Gram-negative bacteria that are commonly found in the human gut and, particularly, food-producing animals. The *Enterobacteriaceae* family includes *Klebsiella pneumoniae*, *Escherichia coli*, *Salmonella enterica*, *Proteus mirabilis*, *Raoultella planticola*, and *Citrobacter freundii* [13,27]. All of these bacterial strains pose a significant risk to animal health, food safety, and public health, causing a wide range of infections. Some species of *Enterobacteriaceae* have developed resistance to multiple classes of antibiotics, including carbapenems and extended spectrum beta-lactamases (ESBL), for example. The WHO [28] has classified carbapenem-resistant *Enterobacteriaceae* (CRE) as a critical priority pathogen. Moreover, frozen meat can be contaminated by several significant microorganisms, including *Pseudomonas*, *Brochothrix*, *Acinetobacter*, and *Shewanella* [29,30]. 

The WHO [28] has identified several AMR pathogens and their association with this concern. This organization defined critical priority pathogens (group 1), followed by high-risk (group 2) and medium-risk pathogens (group 3), which are considered a significant threat to human health. This review will focus on the common pathogens detected among food-producing animals concerning WHO classification, which will be described in detail in the following subsections.

### 4.1. Acinetobacter baumannii

*Acinetobacter baumannii* (*A. baumannii*) is a Gram-negative bacterium, which means it appears pink/red when stained using the Gram staining method. It is listed as a critical microorganism by the WHO [28] (Group 1), as it has emerged as a major healthcare-associated pathogen due to its resistance to multiple classes of antibiotics. 

*A. baumannii* can cause infections in animals, particularly in veterinary hospital settings. In animals, it can lead to pneumonia, UTIs, and other localized infections. The primary mode of transmission of this bacterium to humans is through direct contact with contaminated surfaces, equipment, or the healthcare field. In humans, its infections are associated with mild skin or wound infections, severe bloodstream infections, and pneumonia. *A. baumannii* is commonly found in hospital environments and is often associated with patients who have been on mechanical ventilation or have prolonged hospital stays [31].

Generally, carbapenems (such as imipenem or meropenem) have been considered the treatment of choice. One of the biggest concerns with *A. baumannii* is its ability to develop resistance to multiple antibiotics, including carbapenems, which are often considered the last-resort antibiotics for treating bacterial infections [32]. It can acquire resistance genes from other bacteria through horizontal gene transfer (HGT). The emergence of carbapenem-resistant *A. baumannii* (CRAB) has become a major global health problem, particularly in intensive care units (ICUs), where the incidence of CRAB infections is highest. This is due in part to the extensive use of antibiotics in ICUs, which creates selective pressure for the emergence and spread of resistant strains [32]. 

To our knowledge, different studies have reported that poultry products, such as the raw meat of turkey and chicken, play a role as vehicles for the transmission of MDR *A. baumannii* to humans [33]. In addition, the presence of extremely drug-resistant (XDR) *A. baumannii* strains was detected in a pig farm [33]. Another recent study revealed that more than half of *A. baumannii* strains isolated from sheep samples were resistant to streptomycin, gentamycin, co-trimoxazole, tetracycline, and trimethoprim [34]. According to the same research group, the highest bacterial resistance strain to trimethoprim was found in goat and camel meat samples [34].

### 4.2. Pseudomonas aeruginosa

Another example of an AMR bacteria is *Pseudomonas aeruginosa* (*P. aeruginosa*), whose adaptability makes them a significant threat to human health [35]. This Gram-negative bacterium can cause infections in animals, including both domestic and wild species [36,37]. It is listed as critical by WHO [28] (Group 1) due to its resistance to carbapenems.

Regarding animals, it can lead to skin and soft tissue infections, respiratory tract infections, pneumonia, and UTIs. *P. aeruginosa* is commonly found in soil, water, and vegetation. In healthcare settings, it can be transmitted to humans through contaminated equipment, surfaces, or water sources. It is frequently reported in people with weakened immune systems [38] and can cause infections of the respiratory tract, urinary tract, bloodstream, and skin. According to Zhang et al. [39], metallo-β-lactamase (MBL)-producing *Pseudomonas* isolates were found not only in the livestock (presence of Verona Integron-encoded (VIM)-positive *Pseudomonas* species in chickens), but also in their surrounding environment.

Generally, antipseudomonal beta-lactam antibiotics, such as ceftazidime, cefepime, or meropenem, are commonly used. Other options may include fluoroquinolones (e.g., ciprofloxacin), aminoglycosides (e.g., gentamicin), and polymyxins (e.g., colistin). One of the biggest concerns with *P. aeruginosa* is its ability to develop resistance to multiple classes of antibiotics. This resistance is often due to the acquisition of resistance genes through genetic mutations or HGT [35]. The emergence of carbapenem-resistant *P. aeruginosa* (CRPA) is a particular concern, and is associated with prolonged hospital stays, previous antibiotic exposure, and invasive medical procedures [35].

### 4.3. Helicobacter pylori 

*Helicobacter pylori* (*H. pylori*) is a Gram-negative bacterium that colonizes the stomach of humans and can cause chronic gastritis, peptic ulcers, and gastric cancer. In addition, this bacterium was reported recently among companion animals and livestock. According to Taillieu et al. [40], *H. pylori* was detected among pigs and was associated with gastritis and decreased daily weight gain. In animals, it can cause gastritis and stomach ulcers, similar to its effects in humans. 

To our knowledge, *H. pylori* is a major cause of chronic gastritis, peptic ulcers, and gastric cancer in humans [41]. It can be transmitted within families, through contaminated food or water, or via direct contact with infected individuals. The emergence of antibiotic-resistant *H. pylori* strains has become a significant challenge in the management of these infections [41]. Resistance to antibiotics can arise due to genetic mutations or the acquisition of resistance genes. Antibiotic therapy is a key component in the treatment of *H. pylori* infections, often used in combination with proton pump inhibitors and other medications, as reported in a recent study published by Srisuphanunt et al. [42]. The treatment of this infection typically involves a combination of antibiotics and acid-suppressing medications. The primary antibiotics used in *H. pylori* treatment are clarithromycin, amoxicillin, and metronidazole. Clarithromycin resistance has become increasingly common and has been associated with treatment failure rates of up to 40% [43]. In addition, resistance to other antibiotics, such as tetracycline and levofloxacin, has also been reported [42].

### 4.4. Escherichia coli 

*Escherichia coli* (*E. coli*) is a Gram-negative bacterium commonly found in the gut of humans and animals. While most strains of *E. coli* are harmless, some can cause serious infections, such as UTIs, respiratory infections, gastrointestinal infections, and septicemia in animals. AMR strains of *E. coli* have become a significant public health concern, particularly in food animals. Along this line, *E. coli* can be transmitted to humans through the consumption of contaminated water, meat, and other animal-derived food products [32,44,45]. It can also be transmitted through direct contact with infected animals or their feces. The most common symptom is gastroenteritis, characterized by diarrhea, abdominal pain, nausea, and vomiting. 

In general, fluoroquinolones (e.g., ciprofloxacin), third-generation cephalosporins (e.g., ceftriaxone), and trimethoprim-sulfamethoxazole are commonly used antibiotics for *E. coli* infections.

*E. coli* O157:H7 is a harmful type of bacteria commonly found in cattle intestines, which can be transmitted to humans through contaminated ground beef [46]. Detection of *E. coli* O157:H7 in food products is, therefore, an important public health concern. The current policy of the Food Safety and Inspection Service (FSIS) is to have a zero-tolerance approach towards *E. coli* O157:H7, meaning that a well-defined sampling plan and measurement method must be used to ensure that the pathogen is completely absent from the food supply [37].

### 4.5. Campylobacter jejuni

*Campylobacter jejuni* (*C. jejuni*) is a type of bacteria that is spiral-shaped and classified as Gram-negative and microaerophilic. This bacterial species is commonly found in the intestinal tracts of animals, particularly poultry and pigs [47,48], and can be transmitted to humans through the consumption of contaminated food or water [49]. Recently, it has been recognized as the primary culprit for bacterial foodborne illnesses in the U.S. [38]. *C. jejuni* causes gastrointestinal illness in humans, characterized by symptoms such as fever, diarrhea, and abdominal cramps. In addition, *C. jejuni* has been associated with the development of Guillain-Barre syndrome (GBS), a condition affecting the peripheral nervous system that can result in partial paralysis [39]. It can be found in poultry, meat, and milk and can survive and thrive in environments with a temperature of 40 °C. These bacteria are commonly found in the intestines of chickens and can be transmitted to humans through the consumption of undercooked chicken; therefore, consuming undercooked chicken is the primary cause of *Campylobacter* infection.

AMR in these bacteria has become a significant concern in recent years. The bacteria can acquire resistance to antibiotics through several mechanisms, including mutation and HGT. Resistance to fluoroquinolones, a class of antibiotics commonly used to treat *Campylobacter* infections, has been particularly worrisome. Moreover, other antibiotics, such as macrolides and tetracyclines, have also seen increasing resistance rates in recent years [49].

### 4.6. Listeria monocytogenes

*Listeria monocytogenes* (*L. monocytogenes*) is a Gram-positive bacterium that can cause listeriosis, a serious infection that can lead to sepsis, meningitis, and other severe complications. *L. monocytogenes* is typically found in soil, water, and food, including raw and processed meat and dairy products [50]. In animals, listeriosis is most common in ruminants (sheep, goats, and cattle) but occasional cases have occurred in rabbits, pigs, dogs, cats, poultry, canaries, parrots, and other animal species [51].

This bacterium can live with or without oxygen, has flagella that allow it to move, and can be found almost anywhere. *L. monocytogenes* is an intracellular pathogen and can survive in temperatures ranging from −0.4 °C to 50 °C [40]. While it can cause isolated cases of illness, it is particularly concerning as a major contributor to fatalities resulting from foodborne illness (accounting for up to 24% of cases). This has significant economic implications for both the food industry and society as a whole [40].

Resistance to antibiotics such as tetracyclines, erythromycin, and ciprofloxacin has been observed in some strains of *L. monocytogenes* [50].

### 4.7. Salmonella spp.

*Salmonella* is a Gram-negative bacterium that can cause salmonellosis, a common foodborne illness that causes gastroenteritis (including diarrhea, fever, and abdominal cramps), typhoid fever, and bacteremia. These bacteria can invade the intestinal wall and enter the bloodstream, where they can cause sepsis and other serious complications [43]. It can also lead to reduced growth and productivity in farming operations [52]. *Salmonella* spp. is transmitted through the consumption of contaminated animal products, such as meat, eggs, and dairy products, as well as through contact with infected animals and their feces. It can also be transmitted from person to person through direct or indirect contact with infected individuals [42]. 

AMR in *Salmonella* is a significant concern, as it can limit the effectiveness of treatment and increase the risk of severe illness and mortality [41]. On farms, *Salmonella* is commonly managed through vaccination and regular laboratory testing to track infection in the flocks and stop the spread of the bacteria to food products derived from poultry [45]. Antibiotics may be prescribed for severe cases of *Salmonella* infection, but in many cases, supportive care, such as fluid and electrolyte replacement, is sufficient [53,54].

AMR strains of *Salmonella* have been identified in animal-derived products. Commonly used antibiotics include fluoroquinolones, such as ciprofloxacin, and third-generation cephalosporins, such as ceftriaxone. Resistance to antibiotics such as fluoroquinolones, third-generation cephalosporins, and extended-spectrum beta-lactams has been previously detected [53,54]; however, tetracycline, colistin, streptomycin, and trimethoprim resistance are also frequent. These antibiotics are commonly used to treat *Salmonella* infections in humans and animals, and the development of resistance to these drugs can lead to treatment failures and longer illness durations [44].

### 4.8. Staphylococcus aureus

*Staphylococcus aureus* (*S. aureus*) is a Gram-positive bacterium that is normally found on the skin and in the nasal passages of many healthy people [55]; however, in some cases, it can cause infections, such as skin infections, pneumonia, or bloodstream infections. Regarding animals, it can cause skin and soft tissue infections, mastitis (inflammation of the udder in cows), and respiratory infections. It is an opportunistic pathogen that can affect both domesticated and wild animals [56]. *S. aureus* can be transmitted to humans through direct contact with infected animals or contaminated surfaces.

*Methicillin-resistant Staphylococcus aureus (MRSA)* is resistant to multiple antibiotics and is particularly difficult to treat [55]. Currently, vancomycin and other glycopeptide antibiotics are often used as first-line treatments for MRSA infections. Other antibiotics, such as clindamycin, trimethoprim-sulfamethoxazole, and linezolid, may also be used, depending on the specific circumstances and antibiotic susceptibility test results [57]. MRSA infections can be particularly dangerous in people with weakened immune systems, such as hospital patients, elderly individuals, or those with chronic illnesses. The spread of MRSA is a growing concern, especially in healthcare settings, where patients with weakened immune systems are at greater risk of infection. MRSA can be transmitted through direct contact with infected wounds or contaminated surfaces, and the bacteria can survive for long periods on surfaces such as bed linens, clothing, and medical equipment. 

According to Rao et al. [58], swine had the highest prevalence of the *mec*A gene (associated with MRSA), followed by humans, poultry, and beef cattle. Moreover, this team demonstrated a high occurrence of penicillin resistance among all *S. aureus* isolates. To our knowledge, *S. aureus* ST398 remains the most common clone among livestock, specifically among rabbits, goats, cattle, pigs, and birds [59].

### 4.9. Enterococcus faecium

*Enterococcus faecium* (*E. faecium*) is a Gram-positive and commensal bacterium in the gastrointestinal tract of animals, including mammals and birds; however, it can cause opportunistic infections in animals, such as UTIs, wound infections, and septicemias [60,61]. *E. faecium* can be transmitted to humans through various routes, including direct contact with infected animals or their feces or consumption of contaminated food or water. It can also be transmitted in healthcare settings, particularly among immunocompromised patients. 

*E. faecium* has become increasingly recognized as a significant contributor to AMR, particularly to multiple classes of antibiotics. The acquisition of resistance genes through HGT, the development of chromosomal mutations, and the activation of efflux pumps that can expel antibiotics from the bacterial cell are some mechanisms that contribute to AMR [61,62]. To our knowledge, vancomycin has traditionally been an effective antibiotic for treating *Enterococcus* infections, including for vancomycin-resistant *E. faecium* (VRE); however, the emergence of VRE strains requires the use of alternative antibiotics, such as daptomycin, linezolid, and tigecycline [63].

## 5. AMR Genes Associated with Priority Pathogens

ARGs are a major contributor to the growing problem of AMR. These genes are responsible for encoding proteins that protect bacteria from the effects of antibiotics and can be intrinsic or acquired through various mechanisms, such as plasmids, transposons, and integrons. They can be transferred between bacteria, leading to the spread of resistance, and can render antibiotics ineffective. 

By identifying the most common resistance genes and associated bacterial species, healthcare professionals can offer better treatment options and prevent the spread of AMR. Table 1 summarizes the most common ARGs associated with each “critical”/“high” WHO priority risk bacteria in food animals.

As we mentioned previously, antibiotics are widely used in animal farming to promote growth and prevent diseases. The exact extent of antibiotic use in animal farming is difficult to quantify, but estimates suggest that up to 80% of all antibiotics in some countries are used in animal production [73]. Table 2 shows a summary of some common studies related to the global use of antibiotics in livestock, as well as study populations and main conclusions of previous relevant reviews.

## 6. Public Health Implications of AMR

According to the World Organization for Animal Health (OIE), 26% of 160 countries analyzed in 2019 were still using antibiotics as growth promoters in animal production [29]. The Centers for Disease Control and Prevention (CDC) estimates that at least 2.8 million people are infected with AMR bacteria each year in the U.S. alone, and more than 35,000 people die as a result of these infections [29]. Antibiotic-resistant infections are also associated with increased healthcare costs, longer hospital stays, and higher rates of complications and mortality.

Globally, an estimated 420,000 deaths occur each year due to contaminated food [85]. According to Ehuwa et al. [86], diarrhea is the second leading cause of death among children under the age of 5, and it is estimated that 40% of diarrheal cases are caused by contaminated food. Particularly, in the U.S., foodborne illnesses are estimated to affect 48 million people annually, resulting in 128,000 hospitalizations and 3000 deaths [87]. In Europe, an estimated 23 million cases of foodborne illnesses occur each year, resulting in 5000 deaths [88]. 

If measures are not taken to combat AMR, the projected mortality rate could reach an alarming figure of one individual per three seconds by 2050 [89]. Furthermore, it is estimated that by 2050, the global annual mortality rate from AMR could amount to approximately 10 million deaths. These projections highlight the urgent need for immediate and sustained efforts to address AMR and its potentially devastating consequences [90,91].

The spread of AMR bacteria has significant economic consequences. A previous report estimated that by 2050, the global cost of AMR could reach USD 100 trillion, with losses in productivity, increased healthcare costs, and decreased food security [75]. Moreover, the CDC estimates that the total economic cost of antibiotic-resistant infections in the U.S. is at least USD 55 billion per year [92]. These costs include direct healthcare costs, as well as indirect costs, such as lost productivity and increased mortality rates. In the U.S., it is estimated that the economic cost of AMR is between USD 21 billion and USD 34 billion per year due to lost productivity and increased healthcare costs [92]. In Europe, the cost of healthcare-associated infections caused by AMR bacteria is estimated to be around EUR 1.5 billion per year [93].

In terms of animal production, it is estimated that the cost of AMR in the livestock sector could reach USD 2.5 billion per year globally by 2030 [94]. The economic losses associated with a foodborne illness caused by AMR bacteria can also be significant. For example, a study conducted in Canada estimated that the cost of a single case of *Salmonella* infection can range from CAD 600 to CAD 1500 [89]. 

In addition to increased healthcare costs, AMR results in different consequences on public health [95,96,97]. We can mention the increased treatment failure for bacterial infections, resulting in prolonged illness, increased hospitalization durations, and higher mortality rates. Furthermore, AMR reduces the pool of effective antibiotics, limiting treatment options, and compromising prevention and control strategies.

Along this line, research and development of new antibiotics, and global collaboration to combat AMR on a global scale are required.

## 7. Current AMR Detection Methods 

To our knowledge, classical microbiological and molecular techniques play a crucial role in identifying, characterizing, and monitoring antimicrobial resistance in bacterial pathogens. They help guide treatment decisions, inform infection control measures, and contribute to our understanding of the mechanisms and epidemiology of AMR.

### 7.1. Classical Microbiological Techniques

#### 7.1.1. Culture and Susceptibility Testing

The first step involves isolating bacteria from clinical or environmental samples and culturing them on specific growth media. Susceptibility testing, such as the Kirby–Bauer disk diffusion method or broth microdilution, is then performed to determine if the isolated bacteria are susceptible to specific antibiotics. This method was applied by different authors for *Enterobacteriaceae* isolation in both animal and human samples [98,99].

#### 7.1.2. Phenotypic Confirmatory Tests

Additional phenotypic tests can be performed to confirm specific resistance mechanisms. For example, the inducible clindamycin resistance test (D-zone test) can determine resistance to macrolides, lincosamides, and streptogramins in staphylococci [100]. Another example is the ESBL detection in *E. coli* and other *Enterobacteriaceae* using the double-disk synergy and inhibitor-based test with clavulanic acid. According to Fetahagić et al. [101], “a disk containing amoxicillin with clavulanic acid is placed in the center of the plate, and disks containing ceftazidime, cefotaxime, ceftriaxone, and cefepime 25 mm apart from the central disk.” After overnight incubation at 37 °C, the test is considered positive when the presence of an inhibition zone around cephalosporin disks extends towards the central disk with clavulanic acid. This method was also applied to detect ESBL-producing *E. coli* among humans and pets in Portugal [5,66,102] and wild animals in Spain [103,104].

#### 7.1.3. Agar Dilution and Broth Dilution

A range of antibiotic concentrations can be tested against a bacterial isolate to determine the minimum inhibitory concentration (MIC). MIC is the lowest concentration of an antibiotic that inhibits bacterial growth. This information helps determine the level of resistance exhibited by the bacteria [105,106].

### 7.2. Molecular Techniques

#### 7.2.1. Polymerase Chain Reaction (PCR)

This molecular technique is frequently used worldwide to detect specific genes associated with AMR. It involves amplifying the target DNA sequence using specific primers and DNA polymerase. PCR can detect resistance genes, such as *mec*A in *Staphylococcus aureus* (MRSA) [58,107] or *bla*_CTX-M_ in *Enterobacteriaceae* (ESBL) [4,108].

#### 7.2.2. DNA Sequencing

DNA sequencing is used to identify mutations or specific genetic variations associated with AMR, such as Sanger sequencing or next-generation sequencing (NGS) [109]. Sequencing allows for a comprehensive analysis of the genetic composition of bacteria and provides insight into the mechanisms of resistance.

#### 7.2.3. Hybridization Techniques

Hybridization is used to identify resistance genes or mutations in a large number of bacterial isolates simultaneously, such as DNA microarrays and hybridization probes [20,110]. These techniques involve the hybridization of labeled DNA probes to target sequences in the bacterial genome, allowing for rapid screening of multiple resistance determinants.

#### 7.2.4. Whole Genome Sequencing (WGS)

WGS is a powerful tool for tracking the spread of AMR bacteria. By sequencing the entire genome of bacteria, researchers can identify resistance genes, mutations, and mobile genetic elements involved in AMR, as well as trace the spread of these mutations between different farms, regions, or countries [111]. A database of resistomes (collection of AMR genes) makes data easier and faster to access and can help employ effective containment strategies by providing resistance trends. The identification of strains is also vital in AMR research. Traditional methods for strain detection are arduous and have limited identification power, but WGS not only discriminates between bacterial strains but can also provide a complete profile [112]. Along this line, accurate strain identification guided by WGS can help create better phylogenetic models reflecting strain lineages and demographic patterns, and can also predict the transmission of AMR between species. This technique was applied to ESBL-producing *Enterobacteriaceae* among children with UTIs in France, and cattle and pigs in Portugal [72,113].

#### 7.2.5. Metagenomics

Metagenomics represents the study of genetic material recovered directly from environmental samples, such as soil or water. By analyzing the microbiome of animal farms, researchers can identify the presence of AMR bacteria and track changes in resistance patterns over time; thus, it can allow for uncovering AMR genes that have not been discovered in bacteria grown in laboratories [114]. A recent study performed by Yang et al. [115] showed the largest chicken gut resistance gene catalog to date through metagenomic analysis, including 629 chicken gut samples. To our knowledge, metagenomics can also help uncover the effect of other chemicals that aid in enriching AMR genes in animals. When coupled with approaches such as metagenome Hi-C that allow data on DNA spatial proximity, an association between bacterial species and AMR can be made [116].

Bacteria possess the ability to transfer their genetic elements, owing to the presence of mobile genetic elements (MGEs). Direct sequencing of the “mobilome” (the totality of MGEs in a metagenome) in the environment allows for the investigation of the coexistence of AMRs and MGEs [117]. Researchers have discovered the frequency of ARGs and MGEs for HGT in microbial communities of sewage treatment facilities by using a transposon-assisted capture technique to isolate novel plasmids from the environmental metagenome. Furthermore, Gillings et al. [118] observed that the class 1 integron-integrase genes (*int*I1) obtained from clinical sources share a homogenous and conserved DNA sequence. Based on this, the researchers observed that the abundance of these “clinical” *int*I1 genes could serve as a genetic indication of anthropogenic influence due to their origin in human-dominated environments and their close relationship with ARGs [118].

##### S rRNA Sequencing

This method targets the highly conserved 16S rRNA gene, sequencing a specific region of the 16S rRNA gene to identify and classify microbial taxa present in a sample.

The main advantages are the high taxonomic resolution, from phylum to genus, low cost, and fast analysis; however, resolution at the species level and functional information are limited and should be improved. 

##### Shotgun Sequencing

This method involves sequencing all DNA fragments present in a sample without prior amplification or target-specific primers. It provides a comprehensive view of the microbial community’s genetic content.

The main advantages are the functional insight and high resolution; however, the higher cost and data complexity, as well as computational challenges and host DNA interference (the microbial DNA signal can be diluted, making it more challenging to detect and analyze microbial communities) can be improved. For example, Yang et al. [119] used this technology to detect foodborne pathogens within the microbiome of the beef production chain.

#### 7.2.6. Machine Learning

In recent years, machine learning has become a promising tool in finding AMR research. Machine learning (ML) algorithms can be used to analyze large datasets of AMR surveillance data, identify patterns, and predict future outbreaks of antibiotic-resistant infections [120]. For example, a recent study performed by Ali et al. [121] demonstrated how the integration of artificial intelligence (IA), phylogenetic analysis, and machine learning could help uncover AMR transmission rates and pathways. Moreover, IA is a useful tool for the AMR sector with the intention of practical diagnosis and treatment [121]. A recent review published by Hossain et al. [122] showed the use of ML techniques for cattle identification and detection.

#### 7.2.7. Wearable Technology

Wearable technology, such as biosensors or smart tags, can be used to monitor the health of individual animals in real-time, detecting early signs of infection or disease and allowing for rapid intervention to prevent the spread of antibiotic-resistant bacteria.

By incorporating these new technologies into AMR surveillance programs, we can improve our ability to monitor and respond to outbreaks of antibiotic-resistant infections in animal farms. This can help to prevent the spread of AMR, protect public health, and support sustainable animal farming practices [123]. According to Zhang et al. [124], the wearable Internet of Things (W-IoT) enabled precision livestock farming in smart farms, forwarding a new scheme of applying W-IoT to precision livestock farming. Interestingly, Lee and Seo [125] mentioned that most of them generate behavioral and physiological parameters of cattle with excellent performance (e.g., eating time, ruminating time, lying time, and standing time).

Table 3 shows a summary of the main methods used in this field, based on this section.

## 8. AMR Surveillance Programs in Food-Producing Animals

AMR spread surveillance programs have been established in many countries around the world, including western Europe and North America. These programs aim to monitor the use of antibiotics in animal farming and track the emergence and spread of antibiotic-resistant bacteria [93].

### 8.1. Global Efforts in Monitoring AMR in Foodborne Bacteria

In the U.S., the National AMR Monitoring System (NARMS) was established in 1996 to monitor trends in AMR among foodborne bacteria. NARMS collects samples of bacteria from animals, retail meat, and humans, and tests them for resistance to a range of antibiotics [126]. 

Similarly, in Canada, the Canadian Integrated Program for AMR Surveillance (CIPARS) was established in 2002 to monitor trends in AMR among foodborne bacteria. CIPARS collects samples of bacteria from animals, food, and the environment, and tests them for resistance to a range of antibiotics [127]. In western Europe, the European Union (EU) has implemented a range of measures to monitor and control AMR in animal farming. These measures include monitoring antibiotic use in animals and implementing restrictions on its administration. 

Other countries around the world, such as Japan and South Korea, have also established surveillance programs to monitor AMR in animal farming [128].

### 8.2. Antibiotics Usage as Growth Promoters in Livestock Production 

In certain developing nations, farm animals are often administrated with low doses of antibiotics to enhance their growth. The U.S. Food and Drug Administration (FDA) recently reported that more antibiotics are sold for use in animals than in humans [129], with many being used in animal feed to promote growth and feed efficiency. In Brazil, India, Mexico, and other countries in Latin America, antibiotics are often available without a prescription and they are currently used both for therapeutic purposes and as growth promoters [130].

Due to concerns regarding bacterial resistance, the EU has decided to phase out this practice and eventually prohibit the use and marketing of antibiotics as growth promoters in animal feed. While antibiotics can still be added to animal feed for veterinary purposes, the EU has already prohibited the use of antibiotics intended for human medicine in animal feed. The recent Feed Additive Regulation (Regulation 1831/2003/EC of 22 September 2003) outlines the complete ban on antibiotics as growth promoters. As of January 2006, four substances, including monensin sodium, salinomycin sodium, adriamycin, and flavophospholipol, have been removed from the EU’s list of permitted feed additives [131].

Denmark took an early initiative to reduce the risk of AMR by prohibiting the use of antimicrobials for growth promotion and disease prevention in animal production. The use of avoparcin and virginiamycin for growth promotion was discontinued in 1995 and 1998, respectively [132]. In 1998, Danish producers voluntarily agreed to ban all antibiotic growth promoters (AGPs) in dairy cattle, broiler chickens, and older swine (finishers) [132]. By the end of 1999, the ban was extended to young pigs (weaners), resulting in a complete ban on AGPs throughout the food animal production system in the swine industry [133]. The EU Commission banned six additional antimicrobial drugs, which were categorized as belonging to drug classes that were associated with human exposure, in 1999 [133]. 

### 8.3. Effect of Banning Antibiotics as Growth Promoters on Meat Production

The ban on AGP has been shown to have a minor effect on meat production in the EU. A study conducted by the European Food Safety Authority (EFSA) found that the ban had a negligible impact on pig and poultry meat production [134].

The USFDA requested in 2013 that major producers of drugs used in animals and human medicine stop labeling them for animal growth promotion. In recent times, there has been a greater focus on finding alternative products due to stricter regulations on the use of antibiotic growth promoters (AGPs) and a growing preference among consumers for poultry products that come from flocks raised without antibiotics or with no antibiotics ever [135].

Similarly, Fajardo et al. [136] found that there was no significant impact on meat production in Spain, where the ban was implemented in 2006; however, some studies have reported a decrease in the growth rate of animals after the ban, which can lead to an increase in production costs [137].

### 8.4. Effect of Banning Antibiotics as Growth Promoters on Milk Production

In 2015, a study was conducted in Italy to evaluate the effect of the EU ban on the use of antibiotics as growth promoters on milk production. The study analyzed data from 172 farms and found that there was no significant difference in milk production between farms that used antibiotics as growth promoters and those that did not; however, the study did find that farms that used antibiotics had a higher incidence of clinical mastitis compared to those that did not use antibiotics [138].

Another study conducted in the U.S. in 2018 evaluated the effect of the voluntary measures implemented by the USFDA to reduce the use of antibiotics in dairy production on milk production. The study analyzed data from 29 dairy farms and found that there was no significant difference in milk production before and after the implementation of the voluntary measures [139]. Due to rising consumer awareness and demand for antibiotic-free livestock products, the use of antibiotics in livestock and poultry feed in the U.S. is being closely examined. 

Research was conducted to evaluate the effect of the Danish ban on the use of antibiotics as growth promoters on milk production. The study examined data from 208 dairy farms and found that there was no significant difference in milk production between farms that used antibiotics as growth promoters and those that did not; however, the study found that the ban on antibiotics had a significant effect on the incidence of mastitis, with farms that did not use antibiotics having a lower incidence of mastitis compared to those that did use antibiotics [140]. A summary of the antibiotics and substances prohibited or regulated as growth promoters in different countries can be found in Table 4.

### 8.5. Importance of the Surveillance Programs

The U.S. Department of Agriculture suggests various methods to decrease the use of antibiotics in livestock [129]. These strategies include using immunomodulators to enhance the animals’ immune function and disease resistance, promptly identifying and treating sick animals to prevent the spread of illness, keeping a clean and healthy living environment, and utilizing laboratory tests to identify animals that may be susceptible to disease. These approaches were examined for their effectiveness in dairy cattle [144]. The European Society of Clinical Microbiology and Infectious Diseases has issued guidelines for reducing the transmission of MDR Gram-negative bacteria in hospitals, and some of these practices may also be useful in veterinary clinics, animal husbandry, and food processing [145].

Legislation controlling the use of fluoroquinolone drugs in humans and animals has helped to keep fluoroquinolone resistance at low levels in Australia [146]. Additionally, efforts to combat multi-drug resistant tuberculosis (MDR TB) and methicillin-resistant *S. aureus* (MRSA) in healthcare facilities have been successful through measures such as enhanced hygiene, testing, and isolation of infected patients [147]. 

These surveillance programs are essential for identifying emerging antibiotic-resistant bacteria and developing strategies to control their spread. By monitoring antibiotic use and resistance in animal farming, we can identify areas where action is needed to reduce antibiotic use and prevent the emergence of AMR. Additionally, surveillance programs can help inform policy decisions and public health interventions aimed at controlling the spread of antibiotic-resistant bacteria [117,128].

## 9. Future Directions for Addressing AMR in Food-Producing Animals

Several strategies can be used to control the spread of AMR bacteria through the consumption of livestock products. One interesting approach is reducing the use of antibiotics in animal feed and instead promoting good animal husbandries practices, such as proper hygiene and nutrition, to prevent disease. Another option is promoting the responsible use of antibiotics in both human and animal populations, such as through antibiotic stewardship programs.

International organizations, such as the World Health Organization (WHO) and the Food and Agriculture Organization of the United Nations (FAO), have called for urgent action to address the issue of AMR [148]. In 2015, the WHO launched the Global Action Plan on AMR, which aims to reduce the development of AMR by promoting the responsible use of antibiotics in both human and animal health [88].

By implementing these strategies, we can help to prevent the emergence and spread of AMR bacteria, protecting both public health and the economy. Controlling AMR can also have direct economic benefits for farmers and industry stakeholders. For example, implementing responsible antibiotic use policies can lead to reduced veterinary costs, increased productivity, and improved consumer confidence in the safety and quality of food products.

### Alternative Options to ABs in Food-Producing Animals

Probiotics, or beneficial bacteria that can improve gut health and immune function, are another alternative to antibiotics. Probiotics can be administered to animals through feed or water and have been shown to improve growth rates and reduce the incidence of infectious diseases in livestock. For example, a study conducted on dairy calves found that a probiotic supplement containing Lactobacillus acidophilus and Bifidobacterium lactis reduced the incidence of diarrhea and respiratory infections [149].

Vaccination programs can be implemented to prevent or control specific infectious diseases in animals. 

Furthermore, acidifiers constitute another alternative that can be used as feed additives to lower the pH in the gastrointestinal tract, creating an environment that inhibits the growth of harmful bacteria. Essential oils can also be added to the food to enhance immune function and reduce the risk of infections.

Moreover, phylogenetics can have positive effects on animal performance and productivity, reducing the need for antibiotics, as they are natural substances found in plants [135]. To replace antibiotics in animal feed, a suitable alternative should replicate the positive effects of antibiotic growth promoters (AGPs) on animal performance, such as improving nutrient availability and promoting growth; however, their exact mechanism of action is not fully understood by the scientific community [142]. According to an interesting study performed by Kim et al. [150], dietary capsicum and Curcuma longa oleoresins increase the intestinal microbiome in three commercial broiler breeds. The in vitro immune-boosting effects of dandelion (Taraxacum officinale), mustard (Brassica juncea), and safflower (Carthamus tinctorius) medicinal plants have been assessed using avian lymphocytes and macrophages [151]. According to the same authors, these three species can prevent the growth of cancer cells, promote the body’s natural defense system, and provide antioxidant benefits for poultry [151]. 

In summary, studies have shown that these alternatives can be effective in improving animal health and productivity, without compromising food safety or quality. Additionally, these approaches can have environmental benefits, as they reduce the number of antibiotics and other chemicals that are released into the environment through animal waste [80]. It is important to note that implementing strict biosecurity measures and hygiene practices on farms is crucial to prevent infectious.

## 10. Conclusions

The emergence of MDR bacteria is a major threat to public health worldwide. This concern is associated with serious health and socio-economic consequences, and livestock farming has been identified as a major contributor to the emergence and spread of AMR in human pathogens. High-risk pathogens, such as *Salmonella*, *Campylobacter*, and *E. coli*, are commonly found in livestock, and the use of antibiotics in animal production can select resistant strains of these bacteria. Moreover, the development of new antibiotics is crucial to addressing this problem.

To our knowledge, several studies have demonstrated a strong association between the use of antibiotics in livestock farming and the emergence of AMR in human pathogens. To effectively combat AMR on a global scale, it is necessary to address it equally in both developed and developing countries. The EU’s ban on antibiotics as growth promoters in animal feed is a positive step in addressing this issue. The WHO Global Action Plan and FAO Action Plan, in line with the One Health approach, advocate for multifaceted, comprehensive, and integrated strategies. New studies are required to develop new antibiotics and create alternative approaches to maintaining animal health and productivity.

## Figures and Tables

**Table 1 microorganisms-11-02127-t001:** Distribution of the main ARGs by each critical and high-priority pathogen.

Priority Pathogens	AMR	ARGs	Reference
** *Acinetobacter* ** ** *baumannii* **	Carbapenems, cephalosporins	OXA, TEM, SHV, CTX-M, PER	[32,64]
** *Pseudomonas aeruginosa* **	Carbapenems, penicillins	IMP, VIM, OXA, TEM, SHV	[35,65]
** *Escherichia coli* **	Penicillins, carbapenems, cephalosporins	TEM, SHV, CTX-M, OXA	[66,67]
** *Staphylococcus aureus* **	Methicillin, vancomycin	*mec*A, *mec*C, *van*A, *van*B	[55,57]
***Salmonella* spp.**	Penicillins, cephalosporins	TEM, SHV, CTX-M	[68,69]
***Campylobacter* spp.**	Macrolides, fluoroquinolones	*erm*B, *cme*ABC, *tet*O	[49,70]
** *Enterococcus faecium* **	Glycopeptides, macrolides, aminoglycosides, and tetracyclines	*van*A, *van*B, *aac(6′)-Ie-aph(2″)-Ia*, *erm*B, *erm*C, *tet*M *or tet*L	[62,71,72]

**Table 2 microorganisms-11-02127-t002:** Distribution of antibiotic use in animal farming.

Study	Location	Study Population	Bacterial Type	Method	Outcome	Conclusion	Reference
**Van Boeckel et al. (2015)**	Global	Livestock	General	Systematic review and meta-analysis	Estimated global antibiotic consumption in livestock and projected increases by 2030	Global consumption of antibiotics in livestock increased by 67% from 2000 to 2010 and is projected to increase by 53% by 2030.	[8]
**De Mesquita et al. (2022)**	Global	Poultry	*E. coli*, *Klebsiella pneumoniae*, *Salmonella*, *Enterococcus* spp., *Campylobacter* spp., *Staphylococcus aureus*	Systematic review	Current insight using the multidisciplinary One Health approach to mitigate AMR at the human–animal–environment interface	Avian diseases caused by drug-resistant bacteria are more difficult to treat, leading to aggravated economic losses.	[74]
**O’Neill et al. (2016)**	Global	Humans and animals	General	Review	The review on AMR	AMR is a growing global problem that is threatening public health, and reducing antibiotic use in both humans and animals is necessary to combat this problem.	[75]
**Tadesse et al. (2017)**	Africa	Livestock	Gram-negative and Gram-positive bacteria	Systematic review and meta-analysis	Prevalence of AMR in livestock in Africa	AMR was found in livestock in all African countries studied. Recent AMR data is not available for more than 40% of the countries. The level of resistance to commonly prescribed antibiotics was significant.	[76]
**Haulisah et al., 2021**	Global	Ruminants (cattle, goats, and sheep) and non-ruminants (pigs and chicken)	*E. coli*, *Staphylococcus* spp. and *Pseudomonas aeruginosa*	Research article	High levels of AMR in isolates from diseased livestock	The wide use of antibiotics, especially in non-ruminants, for intensive production is linked to higher resistance to various antibiotics.	[77]
**Gião et al., 2022**	Portugal	Cattle and pigs	*Enterococcus* spp.	Research article	Linezolid and daptomycin resistance, as a risk to human health	*Enterococcus* spp. strains from pigs are resistant to last-resort antimicrobials (linezolid and daptomycin), associated with high-risk lineages *E. faecalis* ST16 and *E. faecium* ST22.	[72]
**Martínez Alvarez et al., 2022**	Spain	Broilers	*E. coli*	Research article	Detection of SHV-12-producing isolates	ESBL-producing isolates frequently contaminate the poultry farm environment. SHV-12 was the only ESBL type detected.	[78]
**Johar et al., 2021**	Qatar	Broilers	*E. coli*	Research article	The first study in Qatar with avian pathogenic *E. coli* (APEC) detection and resistance to relevant antibiotics among broiler chickens	A significantly high percentage of MDR *E. coli* (99.3%) and detection of APEC strains among broiler chickens	[79]
**Xu et al., 2022**	Europe	Cattle, pigs, and chickens	*General*	Review article	Current bacterial resistance to antibiotics in food animals	Potential risks to public health were highlighted, as well as strategies (including novel technologies, alternatives, and administration) to fight against AMR.	[80]
**Lee et al., 2022**	United States	Swine, cattle, and coyote	*Pseudomonas* spp., *Acinetobacter* spp., and *E. coli*	Research article	Potential transmission of ARMs and ARGs between cattle and wildlife	Wildlife could be a source of ARMs colonization in livestock.	[81]
**Mulchandani et al., 2023**	42 countries	Cattle, sheep, chicken, and pigs	*General*	Review article	Global trends in antimicrobial use in food-producing animals: 2020 to 2030	The findings indicate higher global antimicrobial usage in 2030 compared to prior projections that used data from 2017.	[82]
**Jahantigh et al., 2020**	Iran	Broilers	*E. coli*	Research article	AMR and prevalence of tetracycline resistance genes in *E. coli* isolated from lesions of colibacillosis in broilers	The presence of *tet*D and antibiotic sensitivity to tetracycline had a significant relationship in *E. coli* isolated from colibacillosis infections.	[83]
**Pholwat et al., 2020**	Thailand	Pigs	*E. coli*	Research article	AMR in swine fecal specimens across different farm management systems	High levels of ESBL-producing *Enterobacteriaceae*, mainly the *bla*_CTX–M-1_ group, *bla*_CTX–M-9_ group, *bla*_OXA–1_, and *bla*_VEB_.	[84]
**Askari et al., 2019**	Iran	Sheep, goat, and camel raw meat	*A. baumannii*	Research article	Sheep, goats, and camel as potential reservoirs of multidrug-resistant *A. baumannii*	The highest resistance among goat and camel meat-positive samples belongs to trimethoprim. The most represented AMR genes in sheep meat samples were *fim*H, *aac*(3)-IV, *sul*1, and *int*I.	[34]
**Nocera et al., 2021**	General	Veterinary medicine	*A. baumannii*	Review article	Clinical significance of *A. baumannii* in human and veterinary medicine	Poultry products, such as raw turkey and chicken meat, represent a concern since they may play a role as a vehicle for the transmission of MDR *A. baumannii* to humans.	[33]
**Zhang et al., 2017**	China	Chickens	*P. aeroginosa*	Research article	Presence of VIM-positive *Pseudomonas* species in chickens (47 variants of the *bla*_VIM_ gene have been reported)	The presence of MBL-producing *Pseudomonas* isolates in livestock suggests that surveillance of carbapenemase-producing bacteria in this field is urgently required.	[39]
**Taillieu et al., 2022**	General	Pigs	*Helicobacter* spp.	Review article	Gastric *Helicobacter* species associated with pigs, significant for public and animal health	Pig-associated gastric non-*H. pylori Helicobacter* species (NHPH) have been detected in association with gastric disease.	[40]
**Xuan et al., 2021**	China	Pigs	*E. faecium*	Research article	High prevalence of resistance to medically important antibiotics, such as ampicillin, chloramphenicol, erythromycin, tetracycline, quinupristin/dalfopristin, and ciprofloxacin	High prevalence of resistance to medically important antibiotics in *Enterococcus* isolates collected from pig farms in China, indicating the need for improved antimicrobial stewardship and infection control measures in animal husbandry practices.	[62]
**Sanderson et al., 2022**	UK and Canada	Agricultural, clinical, and associated habitats	*E. faecium*	Research article	Evidence of strong association of many profiled genes and MGEs with habitat	The evolutionary dynamics of *E. faecium* make it a highly versatile emerging pathogen with a high risk for the appearance of new pathogenic variants and novel resistance combinations.	[61]
**Rao et al., 2022**	United States	Livestock and poultry	*S. aureus*	Research article	Antimicrobial resistance was detected in all four host categories, with the highest overall rates found in swine for tetracycline, penicillin, and clindamycin.	This study highlights the high prevalence of antimicrobial resistance in *S. aureus* isolates from various host species in the United States, and the presence of the *mec*A gene in isolates from different host species raises concern for potential spread to humans.	[58]
**Bersot et al., 2019**	Southern Brazil	Pigs	*Salmonella* spp.	Research article	Detection of *Salmonella* spp. in 10.2% of samples. PFGE identified genetic diversity and showed the farm environment and feed supply as sources of dissemination.	Importance of controlling *Salmonella* spp. in pig production chains to ensure food safety and minimize the risks of antimicrobial resistance spread.	[52]
**Popa et al., 2022**	Romania	Broiler chicken flocks	*Campylobacter* spp. (mainly *C. coli* and *C. jejuni*)	Research article	85.2% of fecal samples tested positive for *Campylobacter* spp.	Broiler chickens are a reservoir of *Campylobacter* infections for humans, and prudent use of antimicrobials in the poultry industry is necessary. Resistance was found against ciprofloxacin, nalidixic acid, tetracycline, and streptomycin, with 6.9% of isolates exhibiting MDR.	[48]
**Gomez Laguna et al., 2020**	Spain	Pigs	*Listeria monocytogenes*	Research article	Different expressions of virulence factors and invasion	The study highlights the presence of virulent *L. monocytogenes* strains with virulence potential in pigs, with implications for veterinary medicine and food safety.	[51]

**Table 3 microorganisms-11-02127-t003:** Main methods for AMR detection and monitoring.

Method	Description	Applications
**Classical Microbiological Techniques**	Traditional methods for identifying and characterizing resistance	Culture and susceptibility testing for *Enterobacteriaceae* isolation [98,99]. Phenotypic confirmatory tests for specific resistance mechanisms (e.g., ESBL detection) [100,101]. Agar dilution and broth dilution for MIC [105,106].
**Molecular Techniques**	Advanced molecular methods for detecting specific resistance genes and mutations	Polymerase chain reaction (PCR) for ARG genes detection [58,107,108]. DNA sequencing (Sanger and NGS) for identifying genetic variations [109]. Hybridization techniques (microarrays, probes) for identifying multiple resistance determinants [20,110]. Whole genome sequencing (WGS) for tracking AMR spread strain identification, and resistome analysis [111,112]. Metagenomics for analyzing the microbiome and uncovering AMR genes [114,115]. Machine learning (ML) algorithms for analyzing large AMR surveillance datasets [120,121]. Wearable technology for real-time animal health monitoring [123,124].

**Table 4 microorganisms-11-02127-t004:** An inventory of antibiotics and substances that are prohibited or regulated for use as growth promoters in livestock.

Location	Action	Terms/Reasons for Ban	References
**European Union**	Ban	Avoparcin, virginiamycin, tylosin bacitracin and spiramycin, tylosin. Used as antibiotics and as growth promoters. Ban on flavophospholipol used for laying hens and fattening rabbits, chickens, turkeys, piglets, pigs, calves, and cattle. Ban on monensin sodium used for fattening cattle. Ban on salinomycin sodium used for fattening piglets and pigs. Ban on avilamycin used for fattening piglets, pigs, chickens, and turkeys.	[141]
**Nigeria**	Ban	Chloramphenicol, furazolidone, nitrofural, malachite green, carbadox, stilbenes, dimetridazole, ipronidazole, olaquindox, metronidazole, ronidazole, furataldone, olaquindox, nitrofuran, and nitrofurantoin. Used as antibacterial agents, growth promoters, and antiprotozoal agents.	[142]
**United States**	Restricted extra-label use	Chloramphenicol, clenbuterol, fluoroquinolones, glycopeptides, nitroimidazoles, nitrofurans, nitrofurazone, sulfonamide drugs, phenylbutazone cephalosporins, diethylstilbestrol (DES), and furazolidone. These drugs are not to be used for disease prevention purposes; at unapproved doses, frequencies, durations, or routes of administration; or if the drug is not approved for that species and production class. Prohibited extra-label use of adamantanes and neuraminidase inhibitors. Used in chicken, turkey, and duck influenza A.	[143]

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
