# Peer review of "The Silent Threat: Antimicrobial-Resistant Pathogens in Food-Producing Animals and Their Impact on Public Health"

_microorganisms, 2023, doi:10.3390/microorganisms11092127_

Round 1
Reviewer 1 Report
The manuscript presents a description of antibiotic resistance. It is a huge, global problem that is constantly growing. The authors indicate that comprehensive and integrated measures are needed to counter the spread of antibiotic resistance.
The manuscript would be much better reviewed if line numbers were included.
The abstract should be written in the passive voice.
Keywords repeat with words appearing in the title of the manuscript. It is better to change the keywords, it will increase the possibility of searching the article in the database.
In the spread of antibiotic resistance, the failure to observe the withdrawal period from the administration of antibiotics is also of great importance. This is also worth mentioning.
The introduction is written correctly, the literature is properly selected, it ends with a clearly stated goal. Overall, the manuscript is well written. It is typical of a review paper.
I recommend the manuscript for further reviews.
Here are some minor notes for improvement:
4. Priority Pathogens in Food-Producing Animals
and – without italics
4.7. Salmonella spp., Table 1, Table 2
spp. - without italics
Table 1 - all abbreviations should be explained below the table
7.2.1.116. S rRNA Sequencing - please correct the chapter title
Author Response
Thank you for your useful comments. We included line pages in all manuscript.
Regarding the abstract, one sentence was rephrased to the passive voice (lines 17-18). The keywords were also updated to the following: “Antibiotics; Livestock; One Health approach; Multidrug resistance; Foodborne; Pathogens; Healthcare costs.” (line 29-31)
The data regarding the spread of antibiotic resistance was updated in lines 51 and 52. Moreover, minor comments were also included as recommended. We did not include the legend of Table 1 as it includes the antibiotic resistance genes, not abbreviations to describe.
Reviewer 2 Report
The authors presented a nice review regarding antimicrobial resistance, this paper is well-organized and of significance to the general audience. I would recommend it for publication if the following issues can be addressed:
1. the line space around section 3 and many other places is not consistent with the whole manuscript. please check and change it.
2. Ref 13. Delete unnecessary coma.
3. Some of the titles from cited references are lower-cased, and some are capitalized. please uniform them.
Author Response
Thank you for the interesting comments, we appreciate them. All recommended changes were applied in the manuscript.
Reviewer 3 Report
I appreciate the effort carried out by the authors to provide as much information as possible regarding the most critical AMR pathogens among food-producing animals and the widespread dissemination of AMR genes in livestock, as well as proposing future strategies to combat this threat.
AMR is one of the biggest challenges in our era with serious health and economic consequences for the society. In the light of these, every information is not only useful but extremely important. The manuscript is well organized, comprehensive for the reader and with a clear aim. The authors do also well in presenting the conclusions of the paper.
However I have some minor comments about the present manuscript:
1. I consider that the paragraph “To our knowledge…prevent diseases [17]” should be rewritten, since is reiterating the same arguments over again. “Most antibiotics used in agriculture, approximately 90 %, are not administered to treat infections” and “only 10 % of antibiotics are applied to treat infections”.
2. Table 2 is interesting and useful for the scope of the paper. On the contrary, Table 3 presents no usefulness for the reader, since all information are already included in the text. Table 3 should be omitted.
3. The 7th part of the paper about current AMR detection methods, although very interesting, is lightly presented and should be enhanced with a Table presenting scientific results about these methods.
4. 8.6 part of the manuscript “Alternative options to ABs in food-producing animals” presents no logical connection with the rest of the 8th part of the paper titled as “AMR Surveillance Programs in food-producing animals”. Authors should consider omitting this part or integrating it in the 9th part of the manuscript about “Future directions for addressing AMR in food-producing animals”.
Author Response
Thank you for the interesting comments. The minor corrections were included in the manuscript.
We rephrased the mentioned paragraph (lines 81-88), avoiding data repetition. Moreover, Table 3 was removed. A new Table 3 was added based on the main methods used in AMR (line 550). According to point 4, the authors consider maintaining the content of subsection 8.6 and changing it to subsection 9.1. line (678)